# Short Linear Motifs in Colorectal Cancer Interactome and Tumorigenesis

**DOI:** 10.3390/cells11233739

**Published:** 2022-11-23

**Authors:** Candida Fasano, Valentina Grossi, Giovanna Forte, Cristiano Simone

**Affiliations:** 1Medical Genetics, National Institute of Gastroenterology-IRCCS “Saverio de Bellis”, Castellana Grotte, 70013 Bari, Italy; valentina.grossi@irccsdebellis.it (V.G.); giovanna.forte@irccsdebellis.it (G.F.); 2Medical Genetics, Department of Precision and Regenerative Medicine and Jonic Area (DiMePRe-J), University of Bari Aldo Moro, 70124 Bari, Italy

**Keywords:** short linear motifs, protein–protein interactions, SLiM-based small molecules, colorectal cancer interactome, targeted therapy, cancer driver protein interactome

## Abstract

Colorectal tumorigenesis is driven by alterations in genes and proteins responsible for cancer initiation, progression, and invasion. This multistage process is based on a dense network of protein–protein interactions (PPIs) that become dysregulated as a result of changes in various cell signaling effectors. PPIs in signaling and regulatory networks are known to be mediated by short linear motifs (SLiMs), which are conserved contiguous regions of 3–10 amino acids within interacting protein domains. SLiMs are the minimum sequences required for modulating cellular PPI networks. Thus, several in silico approaches have been developed to predict and analyze SLiM-mediated PPIs. In this review, we focus on emerging evidence supporting a crucial role for SLiMs in driver pathways that are disrupted in colorectal cancer (CRC) tumorigenesis and related PPI network alterations. As a result, SLiMs, along with short peptides, are attracting the interest of researchers to devise small molecules amenable to be used as novel anti-CRC targeted therapies. Overall, the characterization of SLiMs mediating crucial PPIs in CRC may foster the development of more specific combined pharmacological approaches.

## 1. Introduction

Colorectal cancer (CRC) is mainly caused by abnormal proliferation of glandular epithelial cells in the colon. CRC is classified into three types: sporadic, hereditary, and colitis-associated [1]. About 65% of patients with CRC have no family history or increased risk of germline mutations and develop the disease due to acquired somatic genomic and/or epigenetic changes [2,3]. The remaining cases of CRC are linked to hereditary factors such as family history (25%), hereditary cancer syndromes (5%), low-penetration genetic variants in some known CRC genes (1%), and other unknown inherited genomic abnormalities [2,4]. Based on current cancer epidemiology, CRC is the third-most commonly diagnosed and the second-most deadly cancer in the world [3]. In 2020, 1.93 million cases of CRC were recorded worldwide, and CRC accounted for 9.4% of all cancer-related fatalities (WHO, https://www.who.int/news-room/fact-sheets/detail/cancer; accessed on 19 July 2022). Moreover, it is predicted that the global incidence of CRC will more than quadruple by 2035, with the greatest increase occurring in less developed countries due to the growing number of cases detected in the older population [1,5].

From a molecular perspective, CRC initiation and progression are complex multistage processes involving a dense network of protein–protein interactions (PPIs). These PPIs mediate cell-signaling cascades that are deregulated as a result of CRC-related alterations [6].

## 2. Short Linear Motifs and Their Emerging Role in Cell Biology and Cancer

It has long been believed that PPIs were mediated by large, structured domains [7,8]. Nonetheless, it is presently evident that a wide variety of protein motifs showing different levels of flexibility are involved in these interactions. Indeed, binding interfaces range from rigid globular domains to disordered regions intrinsically lacking structure [9]. Since organism complexity seems to correlate with a higher degree of disorder in major hub proteins, disordered regions are currently believed to play a functionally relevant role in cell biology [9].

PPIs involved in signaling and regulatory networks are now known to be mediated by an important subclass of disordered interfaces termed short linear motifs (SLiMs). SLiMs are typically conserved, contiguous sequences consisting of about 3–10 amino acids within interacting protein regions [10,11,12]. These motifs have been extensively investigated in the immunological field to define the microbial/host cross-talk underlying immunosurveillance against infectious diseases and some cancer-related oncovirus infections [13,14]. However, SLiMs have aroused growing interest in cancer research in the last decade because they modulate crucial PPI networks involved in the development and progression of many tumors [15]. The SLiMs identified in this field show more variable length and homology sequence levels than those previously characterized, a difference that has been the subject of controversial debate [16]. In addition to being PPI mediators, SLiMs can also function as sites for post-translational modifications, determinants of sub-cellular localization, and targets for proteolytic cleavage [12].

SLiMs are the minimum sequence determinants that can finely modulate cellular PPI networks independently of the wider sequence/structure context where they function [11]. Frequently, SLiMs correspond to consensus sequence patterns located in the disordered regions of proteins, where they mediate transient and low-affinity interactions with various interactors due to their structural flexibility [17,18]. This makes them ideal to mediate dynamic processes such as cell signaling. Consequently, SLiMs play a significant role in determining the spatio-temporal behavior of protein interaction networks [7,19]. In certain instances, a single linear motif is sufficient to mediate an interaction; nonetheless, cooperation between multiple motifs is typically necessary. Cooperation does not require physical contact between the linear motifs and usually depends on two (or more) moderate affinity contacts that synergize to produce a greater effective affinity in the interaction [11]. A significant challenge in SLiM characterization is that, despite their stability in various proteins, SLiMs can show multiple three-dimensional structures [20,21]. In native proteins, SLiMs are intertwined and overlapped to act as functional units. This variety of structural conformations for a single sequence explains why the same unit is sometimes observed in numerous unrelated non-homologous proteins. As a result, proteins may have similar functions despite showing widely different amino acid sequences and three-dimensional structures [21]. On the other hand, the majority of protein domains maintain distinct patterns of amino acid conservation, indicating that they can bind SLiMs with high intrinsic specificity, thereby modulating PPIs [22].

From a molecular point of view, cancer-causing mutations are typically thought to affect protein functionality due to the disruption of their folded globular structure. Nevertheless, 22% of human disease mutations occur in intrinsically disordered regions (IDRs) of proteins [23]. Therefore, alteration of IDR structures is implicated in numerous human diseases, including cancer. The functional role of IDRs is mainly attributable to SLiMs, but the contribution of SLiM defects to cancer pathogenesis remains unclear [11,23]. An interesting computational study including a proteome-wide comparison of the distribution of missense mutations from human disease and nondisease datasets revealed that in IDRs, aberrant mutations are more frequent within SLiM sites [23]. In addition, pathological mutations often affect functionally crucial residues within SLiMs, altering their biochemical properties and interfering with their physical interactions [23]. The analysis of these mutations yielded an exhaustive list of experimentally validated or predicted SLiMs that are disrupted in disease [23]. Compared to mutations in globular domains, the contribution of SLiM mutations to cancer disease may appear to be minor at this time. Nonetheless, emerging data on mutations in predicted SLiMs suggest that this contribution may have been underestimated [23]. In particular, growing findings indicate that a greater emphasis on SLiMs in the coming decades will improve our understanding of cancer-related networks and could be very useful for the development of targeted antitumor treatments [11,23]. Cancer is primarily caused by alterations in cell signaling pathways. Such alterations can occur in IDRs and may disrupt SLiMs, thereby contributing to carcinogenesis due to protein misrecognition or cell-signaling deregulation [23,24].

For instance, several studies have shown that SLiMs play an important role in the activity of BRCA1, a critical tumor suppressor regulating cell cycle progression and DNA-damage-induced checkpoint activation. Two regions (1648–1723; 1756–1842) of the BRCA1 C-terminus (BRCT) domain interact with the _989_TSPTF_993_ BRIP1 (Fanconi Anemia Group J Protein) motif. Alterations in these SLiMs that mediate BRCA1–BRIP1 interaction have been associated with various cancer types, including breast cancer (BC), ovarian cancer, and pancreatic cancer [25,26,27,28,29].

Emerging evidence also suggests that specific SLiMs derived from or encompassing the alpha-fetoprotein (AFP), a well-known embryo-specific and cancer-related protein, may have antitumor effects in several types of cancer (BC, prostatic cancer, and hepatocellular carcinoma) [30,31,32]. In particular, the AFP C-terminal SLiM E_489_MTPVNPGV_497_ can inhibit mouse uterine cell proliferation and has shown anticancer effects in MCF-7 breast cancer cells [30]. In addition, local alignment analysis revealed high similarity between this AFP C-terminal SLiM and AXIN 2, a protein involved in the WNT pathway, which is deregulated in CRC and hepatocellular carcinoma [21,32]. In addition, the N-terminal heptapeptide of AFP (AFP14-20) interacts with the consensus motif (CxxGY/FxGx) of EGF family proteins, including the well-known cancer-related factor EGF [33].

Interestingly, MDMX-mediated pharmacological inhibition of the tumor suppressor P53 is also dependent on a SLiM-related mimicry effect [34]. MDMX, a p53 inhibitor, is regulated by multiple stress signaling pathways. Chen et al. identified an MDMX intramolecular interaction that mimics the interaction with p53, resulting in MDMX autoinhibition. This mechanism involves a hydrophobic peptide located in a long MDMX disordered segment, whose sequence is similar to the p53 transactivation domain [34]. Furthermore, a recent study based on NMR spectroscopy analysis demonstrated that autoinhibition of P53 binding to MDMX requires two SLiMs containing adjacent tryptophan and tryptophan–phenylalanine residues. These SLiMs directly compete for the p53 binding site on MDMX [34,35].

Overall, these data highlight the emerging biological relevance of SLiMs in modulating driver interactions in tumorigenesis and tumor progression, which is further corroborated by the number of different studies and in silico methodologies that have been developed to define a cancer-specific and SLiM-based molecular signature suitable for precision medicine approaches [23,36].

## 3. In Silico Methods to Study PPI Networks

A thorough understanding of PPI networks is crucial for a systems-level interpretation of the vast quantities of molecular data that are made available to researchers. Experimental detection of protein interactions is complicated by a wide variety of binding affinities, and different assays have varying strengths and weaknesses in overcoming these challenges [35].

For instance, based on the evidence that physical interaction between two proteins requires complementary three-dimensional structures, and considering that it is crucial to identify in which conformation a certain interaction can occur, Halakou et al. described a step-by-step method for incorporating the various conformations of proteins into each PPI. In particular, they showed how a PPI network can be refined by docking alternative three-dimensional conformations of each protein participating in binary interactions, thereby unifying the network view and structural perspective of the involved interactions [37].

Computational methods are a powerful tool to bridge the gaps in the experimental approaches used to investigate PPIs. In particular, they can help transfer our understanding of complex interactions from one species to another, integrate our knowledge scattered across databases covering different types of interactions, provide functional insights into the discoveries made from the analysis of omics datasets, and assemble individual interactions into higher-order functional units, i.e., protein complexes and signaling pathways [35].

In this scenario, several in silico strategies using indirect evidence or based on the ability to learn from existing interaction data gathered by experimental techniques, such as two-hybrid and affinity purification coupled with mass spectrometry, have been devised to build screening tools allowing researchers to predict and/or identify PPIs [36,37,38,39]. These tools are useful for a comprehensive interpretation of the enormous amounts of molecular data produced by high-throughput technologies. Current in silico platforms have been developed based on different biological criteria of proteins, such as similarity of amino acid sequences, evolutionary homology profiling, and comparative analysis of protein domain annotations to infer PPIs from shared functional units and similar domain composition, etc.

The above-described strategies provide a huge amount of data and information on PPIs in different species, which are annotated in specific databases. As a result, our current understanding of PPIs is fragmented across multiple sets of data that cover a variety of interaction types across multiple species. The ability to systematically query the databases of interest is crucial for the incorporation of PPI data into effective downstream analysis techniques. Various tools efficiently address this challenge.

The interpretation of high-throughput molecular profiling datasets can be supported by PPI networks. A collection of differentially expressed genes found by RNA-seq, for instance, can have their function clarified by finding enriched biological pathways or by examining their relationships in a larger network context [38,39].

Table 1 below lists various algorithms, servers, and databases that may be useful to researchers to predict in silico novel PPIs and analyze relevant networks to gain insight into their biological functions.

As indicated in Table 1, each tool has distinctive technical purposes that enable a deeper understanding of specific aspects of PPI analysis (e.g., identification of the motifs that mediate binary interactions, three-dimensional structures of binary complexes, evolutionary conservation of PPI motifs, tissue specificity, and particular pathological conditions). Overall, the multitude of available tools highlights the complexity of PPI analysis and the increasing efforts of the scientific community and bioinformaticians toward the development of in silico open-source resources, allowing researchers to study PPI networks in different species and varying pathophysiological contexts.

## 4. In Silico Approaches and Tools to Characterize SLiMs in PPI Networks

As indicated above, SLiMs are short, conserved stretches of amino acids that mediate PPIs. SLiMs are difficult to identify experimentally due to their extremely small size and poor folding. Based on SLiM-recognition domains, numerous computational and bioinformatics tools have been created to analyze PPI networks, especially focusing on cancer-related pathways **[60]**. In the last decade, several in silico algorithms able to extract SLiMs from interaction data have been developed. Surprisingly, emerging evidence suggests that these regions mediate a greater proportion of interactions than was previously anticipated. Moreover, it has been shown that SLiM-mediated interactions can be pharmacologically targeted with specific compounds, which may be used to interfere with or disrupt cancer-related PPI networks [61]. Therefore, several bioinformatics databases and tools have been developed to better characterize and predict the SLiMs involved in PPI networks. A selection of these resources is described in Table 2.

The various in silico resources listed in Table 2 are valuable tools for researchers to gain insight into SLiMs involved in PPIs, but they also have limitations. Since SLiMs are very short peptides, they can be nonspecifically selected by algorithms. Moreover, their limited sequence conservation and loosely folded nature can make detection difficult. Reduced SLiM length also increases the probability of stochastic occurrence of short motifs; hence, the use of pattern matching alone generates a large number of false positive hits [8]. Therefore, methods have been developed to incorporate additional filters based on SLiM characteristics, such as sequence conservation [66,67], structural availability [9,68,69], biophysical feasibility [70], and biological keywords [71].

## 5. SLiMs in CRC Molecular Networks

CRC tumorigenesis is initiated by spontaneous mutations, environmental mutagens, and genetic or epigenetic alterations that trigger the transformation of normal colorectal epithelial cells into tumor cells. Mutations in crucial factors, such as APC and WNT-β catenin (CTNB1) pathway effectors, along with cytokines, chemokines, and growth factors from the tumor microenvironment, cause the hyperproliferation of initiated cells and the subsequent formation of an aberrant crypt and adenoma [6]. Next, the progression of these cells into late adenoma and colorectal adenocarcinoma is caused by mutations in other factors, such as transforming growth factor beta (TGF-β), cell division control protein 4 (CDC4), and SMAD family member 4 (SMAD4) [72]. Lastly, additional mutations targeting major tumor suppressors such as P53 and BAX, anti-apoptotic factors such as BCL-2, pro-angiogenic factors, and extracellular matrix-degrading factors promote CRC cell motility, enabling invasion and metastatization of distant organs [6,73,74].

### 5.1. SLiMs in CRC Signaling Pathways and Tumorigenesis

A few functionally relevant SLiMs have been identified in proteins playing a role in major signaling cascades involved in CRC onset and progression. These include the Wnt/CTNB1, the EGFR/MAPK, and the BCL-2 pathways [35,49,50,57,58].

Wnt signaling hyperactivation promotes tumor cell proliferation and is required for tumor growth [73]. After being secreted and accumulating as a result of Wnt signaling, Wnt ligands bind to Frizzled (Fz) receptors [75]. This leads to the inactivation of the multifunctional glycogen synthase kinase 3β (GSK3β) and to the stabilization, accumulation, and nuclear translocation of CTNB1, which couples with the lymphoid enhancer factor (LEF) or T-cell transcription factor (TCF) and activates specific target genes involved in proliferation and signal transmission [75,76]. Conversely, in the absence of Wnt signaling, CTNB1 is phosphorylated and targeted for ubiquitination and proteasomal degradation by casein kinase 1 (CK1), the APC core proteins, and the axin-GSK3β complex [77].

The proto-oncogene C/EBP homologous protein (CHOP) is a dominant-negative inhibitor of C/EBPs, and its disordered N-terminal region contains crucial SLiMs that are essential for CHOP oligomerization, interactions, and biological activity [78]. In particular, Singh and collaborators described a novel mechanism of CHOP-mediated inhibition of Wnt/TCF signaling and activation of the c-JUN oncogene in HT-29 and DLD-1 CRC cells. This mechanism directly involves the disordered SLiM-containing N-terminal region of CHOP [78]. Further studies will be necessary to identify the exact SLiM sequences involved in CHOP-mediated regulation of Wnt and c-JUN pathways.

Conversely, the CTNB1 SLiM D_32_SGIHS_37_ has been well characterized and is therefore reported in the ELM database (ELM ID: ELMI001302). This motif is directly involved in the aberrant proteasomal degradation of CTNB1 [15,79]. In particular, CTNB1 activates the transcription of Wnt-responsive target genes such as cytochrome C oxidase subunit 2 (COX2), matrix metalloproteinase-7 (MMP7), G1/S-specific cyclin D1 (CCND1), and others [80]. CTNB1 proteasomal degradation begins with GSK3β-mediated phosphorylation of CTNB1 residues S33, S37 (which are part of the above-cited SLiM), and T41. Upon phosphorylation, CTNB1 binds to the E3 ubiquitin ligase β-TrCP through the D_32_SGIHS_37_ SLiM and is routed to proteasomal degradation [79,81]. Point mutations of this crucial SLiM at positions D32, S33, and G34 prevent CTNB1 degradation, promoting its translocation into the nucleus where it functions as a co-activator of Wnt-responsive target genes together with TCF/LEF transcription factors [82].

The ELM database reports other SLiMs in CRC regulatory proteins that are recognized by SCF (Skp, Cullin, F-box) complexes via repeat domains of associated F-box proteins (FBPs) and are therefore routed to subsequent ubiquitin-mediated degradation [15]. For example, the F-box protein family member FBW7 is a ubiquitin ligase that ubiquitinylates various oncoproteins (including G1/S-specific cyclin-E1, MYC, c-JUN, and NOTCH), directing them to proteasomal degradation [83]. The activity of FBW7 is finely regulated by double phosphorylation at conserved TPxxS motifs, which are recognized by various kinases, including cyclin-dependent kinases (CDKs) and GSK3β. GSK3β-mediated phosphorylation links FBW7 activity to the mitogenic signaling pathway [83,84]. CRC is caused by different genomic alterations, such as microsatellite instability, altered CpG islands methylation levels, and more frequently (85% of all cases) chromosomal instability [85]. These alterations can affect the recognition sites of several GSK3β substrates, as detailed in Table 3.

The EGFR/MAPK signaling pathway is also directly related to CRC oncogenic processes and plays a critical role in tumor growth and disease progression [111]. EGFR is a transmembrane tyrosine kinase receptor (RTK) that is activated by the autophosphorylation of several tyrosine residues in its intercellular domain and dimerizes after ligand binding [112]. EGFR–ligand complexes activate the RAS protein, which in turn triggers the MAP kinase cascade and the extracellular signal-regulated kinase (ERK), both these signals being initiated through phosphorylation of specific serine and threonine residues [113,114]. As a result of its involvement in CRC tumorigenesis, this pathway and its downstream signaling cascades have been identified as potential targets for CRC therapeutic strategies [115,116].

The proliferative signal associated with the cascade activation of MAPKs depends specifically on 3 SLiM clusters: two clusters enriched in positively charged amino acids surrounding a central cluster enriched in hydrophobic amino acids. These motifs, consisting of a series of positive–hydrophobic–positive amino acids, represent the modular structure of the docking PPIs involved in MAPK signaling. Therefore, point mutations in these crucial amino acids may alter the activation of MAPK signaling cascade [117].

Apoptosis is a crucial cell-death process that is frequently dysregulated in various malignancies, including CRC [118,119]. Apoptosis is mainly controlled by the BCL-2 protein family, whose members are involved in CRC progression and chemoresistance [74]. Each member of the BCL-2 protein family exhibits homology in one or more of the four BCL-2 homology (BH) domains [74,120]. The anti-apoptotic members BCL-2, BCL-XL, MCL-1, BCL-W, and A1/BFL-1 contain four BH-domains (BH1-BH4). In their tertiary protein structure, the BH1, BH2, and BH3 domains help to create a hydrophobic pocket. The pro-apoptotic members include the BH3-only proteins and the effector proteins. BIM, BAD, BID, PUMA, NOXA, BMF, HRK, and BIK are examples of BH3-only proteins, so termed because they only exhibit homology to the BH3 domain of BCL-2. The effector proteins contain three to four BH domains and comprise BAX and BAK, which can create macropores in the mitochondrial outer membrane, thereby inducing mitoptosis [74].

The BH3 motif has been identified as a relevant SLiM in CRC tumorigenesis [16,74]. It is shared by two major BCL-2 protein subgroups, i.e., BCL-2 homologous proteins, which are evolutionarily related to BCL-2 by common ancestry, and BH3-only proteins, which do not appear to have any evolutionary or structural ties to BCL-2 homologs or to each other [16]. Recent evidence shows that the current models of apoptosis regulation ignore a third, larger group of even more varied BH3-containing proteins [16]. The ambiguous definition of the BH3 motif contributes to the lack of research on the biological effects of this third group and how it affects the BCL-2 protein network. On the other hand, the addition of this third group of proteins complicated the exact definition of the BH3 signature, though the hexameric sequence L-X(3)-G-D is frequently used to refer to this motif [121]. The alignment of the 63 reported BH3 motifs contained in the three subgroups confirms that only two crucial residues (L and D) are conserved among them [16,121,122].

Pathological variants of crucial SLiMs have also been observed in inherited CRC syndromes. In particular, familial adenomatous polyposis (FAP) is an autosomal dominant disorder characterized by a predisposition to colorectal polyposis, which in some cases will evolve into colorectal carcinoma if not surgically treated. A variant of FAP is the Gardner syndrome, in which multiple adenomas of the colon and rectum occur with desmoid tumors and osteomas [95]. This condition can be caused by the point mutation Ser171Ile in the linear motif (Q_163_NLTKRIDSLPLTE_176_) of APC nuclear export signal (NES) [96,97].

### 5.2. SLiMs in CRC Hallmarks: A Case Study

Recent data from our group are consistent with the crucial role played by SLiMs in the PPI network of SET and MYND domain containing 3 (SMYD3) in different cancer study models, including BC, CRC, and other gastrointestinal tumors [123,124]. In the last few years, our laboratory has focused on this methyltransferase, which is overexpressed in many types of human tumors, although its oncogenic role has not been fully understood yet [125]. In particular, we searched for novel SMYD3 interactors involved in cancer-related pathways. To this aim, based on the emerging evidence that tripeptides are the minimum determinants able to mediate PPIs [126,127,128,129], we generated a library of 19 tripeptides (termed P1 to P19) mainly composed of rare amino acids and therefore suitable to serve as valuable minimum PPI-mediating motifs. Indeed, several computational studies confirmed that amino acids occurring less commonly in proteomes (i.e., encoded by 1–3 codons) have a higher biological significance than more frequent amino acids (i.e., encoded by 4–6 codons) in PPI networks [130,131,132]. Thus, we first tested the in vitro binding affinity of tripeptides P1-P19 to SMYD3 and then used them as in silico probes to screen the human proteome in the search for novel SMYD3 interactors [123]. Enrichment of these tripeptides was observed in DNA repair pathway proteins. In particular, this analysis allowed us to identify in silico BRCA2, ATM, and CHK2 as direct SMYD3 interactors. These interactions were subsequently validated in vitro and in cellulo [123]. Furthermore, an in vitro competition assay confirmed the direct involvement of the identified tripeptides in SMYD3 binding to BRCA2 and ATM. Indeed, the purified tripeptides inhibited in a dose-dependent manner the physical interaction between SMYD3 and BRCA2/ATM fragments containing the corresponding tripeptide sequences [123].

Next, to gain insight into novel SMYD3 cancer-related activities, we focused on the whole-proteome distribution of the P1-P19 tripeptides and assessed the biological function of each putative SMYD3 interactor to identify the most important candidates associated with cancer hallmarks [124]. Surprisingly, this computational tripeptide screening of the human proteome allowed us to identify crucial cancer-related proteins such as mTOR, BLM, MET, AMPK, and RBL2 (p130) as novel SMYD3 interactors [124]. Subsequently, these interactions were validated in CRC and gastric cancer cell lines. In particular, the interaction between SMYD3 and AMPK was confirmed in multiple gastrointestinal cancer cell lines (CRC, GC, hepatocellular carcinoma, pancreatic cancer), confirming the role of SMYD3 in the metabolism of gastrointestinal cancer [124].

A significant benefit of this strategy is that our tripeptides can be used to develop pharmacological inhibitors of SMYD3 oncogenic PPIs in order to modify the composition of relevant multiprotein complexes associated with cancer driver proteins. Similar to other approaches devised in the last few years [133,134,135], our strategy provides an appropriate in silico methodology to facilitate the identification of novel interactors and generate small molecules suitable to be used as oncoprotein inhibitors.

This methodology is schematically outlined in Figure 1.

### 5.3. SLiMs in CRC-Related Microbiome

Growing evidence suggests that microbes represent relevant players in CRC tumorigenesis [136]. Thus, the short motifs modulating oncovirus-host PPI networks are an emerging example of SLiMs that may be involved in CRC tumorigenesis and related molecular networks [13,14,137]. Although the mechanisms underlying the interactions between colonocytes and the surrounding environment are not clear yet, the John Cunningham virus (JCV), human papillomavirus (HPV), and Epstein–Barr virus (EBV) have been linked to CRC [13]. Various viruses tend to mimic the SLiMs of host proteins to use cellular processes to their advantage. In particular, they have developed the capacity to interact with elements of the host cell via protein SLiMs that resemble those of the host, which facilitates their internalization and the manipulation of a wide range of cellular networks [137]. This mechanism, known as molecular mimicry, makes these SLiMs potential therapeutic targets [138].

Interestingly, SLiM-based molecular mimicry is investigated as a significant aspect of microbiome-host relationships, which are of crucial relevance in CRC tumorigenesis and inflammatory bowel diseases [139]. In particular, a recent computational study identified six SLiMs that are involved in *F. nucleatum*–human cross-talk in the context of gastrointestinal diseases [139].

Taken together, these data confirm the biological significance of SLiMs in cancer-related pathways and suggest that they may play a role in PPI networks involved in CRC. In addition, they could be used to devise pharmacological strategies to interfere with the main signaling cascades driving tumorigenesis and cancer progression.

## 6. Potential Small-Molecule Anticancer Drugs Based on SLiMs and Short Peptides in CRC: Where Do We Stand?

### 6.1. Pharmacological Suitability of SLiMs and Short Peptides as Anticancer Drugs

The biological relevance of PPIs in key cancer-related processes, including cell growth, proliferation, differentiation, and signal transduction, is corroborated by the evidence that PPIs are altered in various cancer types [140,141]. This provides alternative therapeutic prospects targeting cancer-related PPI networks to disrupt carcinogenesis and cancer progression signaling. Currently, only 1% of all human proteins that are deemed to be druggable have been targeted by small molecules based on SLiMs [142]. Targeting PPIs is more difficult than using conventional drug discovery techniques such as designing small compounds that bind to enzyme active sites.

There are different structural caveats to the pharmacological targeting of PPIs, including their large interfaces, which are frequently flat or shallow. Historically, drug design and discovery have given rise to the belief that PPIs are difficult to target because the compounds should be small enough to enter the cell and also be able to affect the large and often shallow PPI interaction sites [142]. Despite this, several studies demonstrated that protein–protein interfaces or the regions located nearby are often flexible or intrinsically disordered, allowing a small molecule to penetrate these complexes and displace the relevant protein interaction partner [143,144,145,146].

Furthermore, current advances in molecular profiling of tumor samples as a result of high throughput techniques such as NGS and mass spectrometry enable the rapid detection of alterations in genes encoding oncoproteins, providing a large number of potential therapeutic targets and offering opportunities for the design of new drugs. Notably, based on different studies carried out by pharmacological companies and academic groups on SMYD3 cancer-related activity, novel and more efficient SMYD3 inhibitors have been generated [147,148,149].

The development of novel anticancer drugs is complicated by limitations related to the specific chemical characteristics of potential therapeutic molecules and the structural requirements of druggable targets [61]. The chemical qualities required for anticancer drugs are somewhat limited and largely determined by the type of drug administration to the patient [150,151]. In particular, peptide-based drugs have several limitations, especially in the case of short peptides. These limitations range from biological stability against lytic enzymes (such as peptidases) to insufficient pharmacokinetic profiles for oral absorption or other routes of administration [151].

Most drug targets are molecules with binding sites for low-molecular-weight compounds. These compounds are then used as starting points for the development of small chemical analogs that bind to the same site and can function as competitive or irreversible inhibitors [61]. If a peptide shares the same binding characteristics as one of the physiological interaction partners of a protein of interest, it can act as a competitive inhibitor. Two types of peptides can be considered for this purpose: peptides matching one of the two interacting proteins and synthetic peptides chosen from a virtual screening of a peptide library. Usually, competitive inhibitors designed for cancer therapy can mask an interaction domain by making it inaccessible to a natural interaction partner that is required for cancer-related PPIs and phenotypes [152,153,154]. Subsequently, these peptide–ligand interactions involving essential regions of the target protein can be used in high-throughput screenings to identify low-molecular-weight molecules with functionally similar analogs [155,156,157].

As a matter of fact, SLiM-based small molecules are still in their early days. For instance, BH3-mimetics are SLiM-based engineered peptide inhibitors able to bind to the hydrophobic groove of the corresponding anti-apoptotic proteins [16,158]. In particular, in cellulo and in vivo studies demonstrated that the SMAC mimetic JP-1201 reduced HT-29 cell survival and CRC tumor growth through an additive effect on apoptosis and disruption of DNA repair mechanisms, but the underlying molecular mechanisms have not been clarified yet [159]. On the other hand, several short peptides have already shown their potential as novel antineoplastic agents.

### 6.2. Short Peptides as Potential Anti-CRC Drugs

One of the first examples of short peptides tested for their anticancer effects are the parasporin (PS) peptides isolated from the gram-positive bacterium Bacillus thuringiensis [160]. A total of 13 PS proteins have been identified in 11 different strains of B. thuringiensis. Eight of these proteins are included in the PS1 family (PS1Aa1, PS1Aa2, PS1Aa3, PS1Aa4, PS1Aa5, PS1Ab1, PS1Ab2, PS1Ac1), two in the PS2 family (PS2Aa1, PS2Ab1), two in the PS3 family (PS3Aa1, PS3Ab1), and one in the PS4 family (PS4Aa1) [160]. The most investigated anticancer effect is related to PS2, which functions as a cytolysin, permeabilizing the plasma membrane with target cell specificity and then inducing cell death [161]. A recent study revealed the potential of PS2 parasporin family members as anti-CRC drugs. In particular, the PS2 short peptides P264-G274, loop1-PS2Aa, and loop2-PS2Aa displayed high cytotoxicity in SW480 and SW620 CRC cells after 48-h exposure [162]. Parasporin-2Aa1 (PS2Aa1), also known as cry46Aa1, is a protoxin with known anticancer properties, which is generated by B. thuringiensis during sporulation. This 37 kDa toxin is activated by serine proteases such as proteinase K and trypsin, resulting in the production of a highly toxic 30 kDa peptide that is effective against cancer cells [162,163,164]. However, in CRC cells, the exact mechanism of PS2Aa1 activity and the receptors involved in its interaction with cells remain unclear. Another study reports the induction of apoptosis as a mechanism of cell death alongside the inhibition of various survival pathways, including AKT, E3 ubiquitin–protein ligase XIAP (XIAP), ERK1/2, and the activation of the tumor suppressor proteinase-activated receptor 4 (PAR-4) in PC-3 pancreatic and HEPG-2 hepatic cancer cells treated for 24 h with PS2Aa1 [165].

Various studies support the use of short peptide aptamers as promising pharmacological inhibitors of signal transducer and activator of transcription 3 (STAT3). STAT3 is a transcription factor that is activated by tyrosine kinase phosphorylation and has a well-defined domain structure [61,153,156]. Upon phosphorylation by tyrosine kinases activated by ligand binding to growth factor receptors, STAT3 dimerizes and translocates into the nucleus, where it binds to specific DNA response regions and regulates transcription [166]. In most cancer cells, STAT3 is phosphorylated as a result of either increased oncogenic signals or decreased tumor-suppressive pathways. It has been shown that sphingosine-1-phosphate receptor-1 (S1PR1) upregulates tyrosine–protein kinase JAK2 (JAK2) activity, increasing STAT3 phosphorylation in tumor cells, and STAT3 regulates S1PR1 and IL-6, resulting in a positive feedback loop [167,168]. STAT3 activation affects proliferation, apoptosis, differentiation, angiogenesis, immune cell recruitment, and metastasis, all of which are cancer hallmarks [167,168]. Borghouts and colleagues demonstrated that the recombinant STAT3-specific peptide aptamer rS3-PA can inhibit STAT3 and has specific anticancer activity [153]. Peptide aptamers are 12–20-amino acid-long molecules that can be selected from random, high-complexity peptide libraries in yeast two-hybrid screens [152,153]. The authors carried out an in silico virtual screening of a peptide aptamer library to identify competitive inhibitors that could interfere with cancer-related functions. They used human thioredoxin devoid of cysteines as an optimal scaffold for the display of target-interacting peptides in a restricted conformation and generated recombinant proteins for the delivery of specific peptide aptamers to cells [152]; rS3-PA was shown to rapidly enter cancer cells, decrease STAT3 phosphorylation, and increase its proteasomal degradation, promoting cell growth arrest and apoptosis [153]. Further studies will be necessary to assess the in vivo pharmacokinetic and bioavailability characteristics of rS3-PA treatment.

Anticancer peptides (ACPs) are an emerging class of naturally occurring or synthetic anticancer compounds. They show greater selectivity for cancer cells and less propensity for drug resistance [169]. In a recent study, ACP candidates were identified and selected from an in silico pepsin hydrolysate screening of *Cordyceps Militaris* (CM) proteome using various machine learning-based ACP prediction servers, i.e., AntiCP, iACP, and MLACP. The purpose of the authors was to select CM-derived ACPs to be used as an alternative or adjunct treatment for CRC, hence minimizing the need for chemotherapy. To confirm their anticancer effects, CM-biomimetic peptides were tested in vitro in a non-metastatic colon cancer cell line in comparison and in combination with doxorubicin, a typical chemotherapeutic agent for the treatment of colon cancer. Overall, the results demonstrated that the selected biomimetic peptide C-ori improved the efficacy of doxorubicin treatment [170].

Another group performed an in silico analysis to generate a new library of short cationic amphiphilic α-helical ACPs with specific cytotoxicity against colorectal and cervical cancer [171]. The α-helix is the most prevalent secondary structure shown by ACPs. This study provided important findings on structural and pharmacokinetic features required for peptide-mediated anticancer activity. In particular, the authors found that peptides predominantly consisting of lysine (K) residues (e.g., CIIKKIIKKIIKK-NH_2_) in their hydrophilic domains exhibit more selective anticancer activity, whereas peptides containing arginine (R) (e.g., CIIRRIIRRIIRR-NH_2_) display strong toxicity in normal cells. Furthermore, it was shown that ACP anticancer activity depends on their helical composition and hydrophobicity [171]. Indeed, as confirmed in multicellular CRC spheroid models, the addition of two isoleucine residues at the C-terminus of selected ACPs (e.g., CIIKKIIKKIIKKII-NH_2_) increased their anticancer activity by boosting their hydrophobicity and helical content. The higher ACP toxicity detected in cancer cells compared to normal cells was due to better penetration into the negatively charged cancer cell membranes, resulting in greater cellular uptake, and their cytotoxic effect was primarily exerted by damaging mitochondrial membranes, resulting in apoptosis [171].

Overall, these data confirm that short peptides and SLiMs exhibit promising pharmacokinetic characteristics in vivo.

### 6.3. Short Peptides in Clinical Studies

Several short peptides have been recognized as PPI modulators and therefore investigated in clinical trials for CRC treatment (Table 4), which reveals an emerging interest in transferring knowledge about these compounds into clinical applications.

A major advantage of short peptides is their high targeting specificity; however, the number of terminated clinical trials suggests that new nanotechnologies are needed to enhance the chemical, physical, and biological stability of these compounds [151].

New therapeutic opportunities may come from highly specific small-peptide inhibitors called BH3 mimetics, which have been developed to target anti-apoptotic BCL-2 proteins by mimicking the action of BH3-only proteins [74]. By interfering with the interaction between BH3-only and prosurvival BCL-2 family proteins, these SLiM-based molecules may eventually promote cancer cell death [16,74,120]. Several BH3 mimetics-based pharmacological compounds (i.e., Venetoclax, Navitoclax, AZD5991, AMG-176, S64315, and others) are being studied in ongoing clinical trials for the treatment of leukemia and hematological malignancies (Clinical trial Ids: NCT03592576, NCT03181126, NCT03218683, NTC03797261, NTC03672695, NTC02979366; https://clinicaltrials.gov/; accessed on 5 September 2022) [172]. Notably, clinical trials on Navitoclax have been extended to solid tumors (NCT03592576 NCT03181126; https://clinicaltrials.gov/; accessed on 5 September 2022) [172]. Thus, given the involvement of BCL-2 proteins in colorectal tumorigenesis, it would be interesting to also investigate BH3 mimetics in CRC [172].

## 7. Conclusions

Currently available treatments for colon cancer are based on specific combinations of surgery, radiation, chemotherapy, and targeted therapy. Nevertheless, following surgical resection and intensive chemotherapy, 50 percent of CRC patients have disease recurrence. In addition, the occurrence of chemoresistance hampers the efficacy of chemotherapeutic agents [173]. Therefore, there is an urgent need for the development of safer and more selective anticancer drugs with novel modes of action.

Moreover, it is crucial to improve the clinical management of CRC patients in order to prevent relapse. This should be combined with the search for novel and more efficient approaches to obtain additional diagnostic and prognostic information.

An efficient approach to identify cancer drivers, as well as diagnostic markers for accurate tumor staging and prediction of clinical outcomes, involves in silico studies to detect differentially expressed genes across different phenotypes, followed by the analysis of the relevant pathways with specific bioinformatics tools evaluating the enrichment of altered oncoproteins [174].

In the last decade, efforts in cancer research have increasingly focused on the development of integrated strategies combining in silico, in vitro, in cellulo, and in vivo approaches to provide in-depth knowledge of cancer-related pathways, PPI networks, and the biological implications underlying a multifactorial disease such as cancer [175]. Indeed, CRC initiation and progression are complicated multistage processes depending on a dense network of PPIs organized in signaling cascades that become dysregulated as a result of abnormalities in various signaling effectors [6].

Recent advances in high-throughput biotechnology applications enable large-scale analysis of disease-associated genes and proteins involved in critical molecular processes. The biological and clinical significance of the resulting data needs careful evaluation. Characterization of the critical SLiMs that mediate PPIs between key players in CRC tumorigenesis and progression is expected to support the development of more efficient and specific pharmacological approaches based on small-molecule anticancer drugs, which may be used in combination with chemotherapeutic agents or pharmacological inhibitors (dual targeting, synthetic lethality, and co-targeting strategies). For instance, the association between pharmacological inhibitors such as ralimetinib (a MAPK14 inhibitor) and SLiM-based small molecules interfering with MAPK signaling activation may prove an effective therapeutic approach to hamper MAPK-dependent CRC development and progression [117]. Furthermore, SLiMs represent promising tools to develop novel immunotherapeutic strategies such as CRC vaccines to prevent oncovirus-dependent colorectal tumorigenesis (JCV, HPV, and EBV).

Further studies are required to gain deeper insight into the therapeutic potential of SLiM-based small molecules in CRC and to address the main limitations related to their stability, bioavailability, and pharmacokinetics.

## Figures and Tables

**Figure 1 cells-11-03739-f001:**
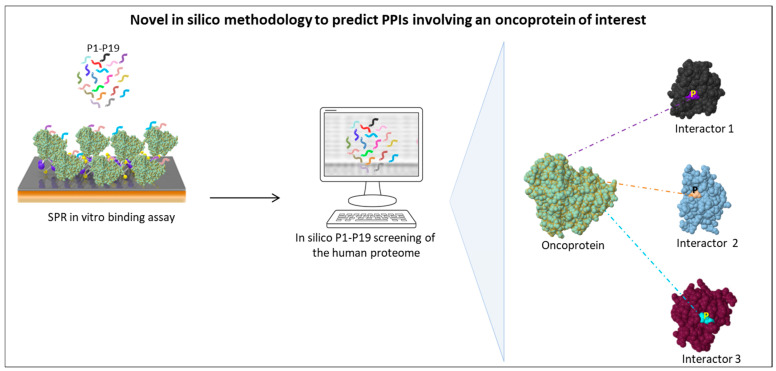
Schematic representation of a novel in silico methodology developed by our group to search for new interactors of an oncoprotein of interest by taking advantage of a library of SLiMs (tripeptides P1–P19) able to bind it in vitro, as identified by surface plasmon resonance (SPR) analysis [124]. Tripeptides P1–P19 are then used as in silico probes to identify human proteins containing them, which are therefore candidate interactors of the oncoprotein of interest.

**Table 1 cells-11-03739-t001:** List of relevant in silico tools used to predict and analyze PPIs.

Name of In Silico Resource	Website	Description	Technical Advantages	Refs
PINA (Protein Interaction Network Analysis)	https://omics.bjcancer.org/pina/ (accessed on 3 September 2022)	An integrated server for PPI non-redundant and curated data referred to six model organisms	A useful tool to provide comprehensive PPI information through integrated visualization of PPI data and construction, filtering, and data analysis	[40,41]
SPRINT (Scoring Protein INTeractions)	www.csd.uwo.ca/faculty/ilie/SPRINT (accessed on 3 September 2022)	This algorithm enables the computational organization of existing PPI networks to the level of species interactomes	It simplifies the interpretation of the results and makes them more objective through a PPI score	[42]
Path2PPI	http://bioconductor.org/packages/release/bioc/html/Path2PPI.html (accessed on 3 September 2022)	This algorithm analyzes the homology between protein sequences of multiple organisms or a single target organism.	It allows users to combine sequence similarity searches of the examined proteins with their functional information about a particular pathway	[43]
Paralog Matching	https://github.com/Mirmu/ParalogMatching.jl (accessed on 3 September 2022)	This algorithm predicts interacting paralogs between two distinct protein families. It enables homology analysis of all members of two protein families that belong to the same species and predicts PPIs, maximizing the detectable coevolutionary signal.	It provides a direct correlation by amino acid occurrences between multiple sequence alignment (MSA) and interprotein residue–residue contacts in the PPI	[44]
RaptorX-ComplexContact server	http://raptorx.uchicago.edu/ (accessed on 3 September 2022)	This server analyzes the interfacial contacts between two potentially interacting heterodimeric protein sequences using deep-learning techniques.	A useful tool for protein docking analysis, protein–protein interaction prediction, and protein interaction network construction	[45]
COZOID (Contact Zone Identifier)	http://decibel.fi.muni.cz/cozoid (accessed on 3 September 2022)	This algorithm analyzes several docking structures covering the three major types of PPIs (coiled-coil, pocket-string, and surface–surface interactions) and their contact zones with different levels of detail.	It provides docking models of interacting proteins and enables the selection of the best docking structures based on their similarity to a conserved structure from reference homologous proteins	[46]
Path-LZerD	https://kiharalab.org/proteindocking/pathlzerd.php (accessed on 3 September 2022)	This software predicts the assembly order of multimeric proteins starting from single subunit structures.	A useful tool to design drugs that target crucial interactions within a specific complex	[47]
ReactomeFIViz	https://reactome.org/tools/reactome-fiviz (accessed on 3 September 2022)	This Cytoscape application facilitates the pathway- and network-based analysis of RNA-seq and other omics datasets using the Reactome pathway database.	It allows users to link the PPI dataset reported in two or more databases	[48]
KeyPathwayMiner	https://apps.cytoscape.org/apps/keypathwayminer (accessed on 3 September 2022)	This Cytoscape application detects highly connected PPI networks in which genes show similar expression.	A useful tool to combine interaction network data with omics datasets in order to identify novel functional peptide modules	[49]
VieClus (Vienna Graph Clustering)	http://vieclus.taa.univie.ac.at/ (accessed on 3 September 2022)	This software enables the visualization of PPI clusters showing similar functional modules.	It allows users to identify functional modules by searching for sets of proteins whose interactions are dense within the sets but sparse between the sets	[50]
TD-WGcluster (Time Delayed Weighted Edge Clustering portal)	https://www.r-project.org/ (accessed on 3 September 2022)	This algorithm integrates the three-dimensional topology of PPIs.	It combines PPI topology with a dynamics component derived from time series data	[51]
SANA (Simulated Annealing Network Aligner)	https://sana.ics.uci.edu/ (accessed on 3 September 2022)	This alignment software compares PPI motifs between different species.	A useful tool to perform comparative analyses of PPI networks to reveal evolutionary relationships between species	[52]
PEPPI (Predicted Protein-protein Interactions)	https://zhanggroup.org›PEPPI (accessed on 3 September 2022)	This alignment software predicts the exact peptide modules involved in binary interactions between two amino acid sequences.	It integrates multiple independent prediction and analysis methods of protein sequence similarity, structural homology, functional association, and machine learning-based classification	[53]
CPDB (Consensus PathDB)	http://cpdb.molgen.mpg.de/ (accessed on 3 September 2022)	A comprehensive database for studying human PPI networks and related information (biochemical pathway, genetic, metabolic, signaling data, and drug–target interactions).	A useful tool to provide a correct interpretation of the massive quantities of PPI molecular data	[54]
IID (Integrated Interactions Database)	http://iid.ophid.utoronto.ca/ (accessed on 3 September 2022)	A curated database containing comprehensive information on PPIs detected and predicted in 18 species, including humans.	A useful tool to study PPIs in specific conditions (e.g., tissues, developmental stages), conservation across species, directionality of the interaction, and druggability	[55]
HIPPIE (Human Integrated Protein-Protein Interaction rEference)	http://cbdm-01.zdv.uni-mainz.de/~mschaefer/hippie/ (accessed on 3 September 2022)	This resource integrates multiple human PPI databases.	It allows users to overlay gene expression data and other annotation resources to construct protein networks specific to a tissue, disease, or subcellular localization	[56]
MIPS (Mammalian Protein-Protein Interaction Database)	http://mips.helmholtz-muenchen.de/proj/ppi/ (accessed on 3 September 2022)	A manually curated database of high-quality PPI data from the scientific literature.	A useful tool for the metanalysis of current scientific literature on mammalian PPIs	[57]
OncoPPi Portal	https://oncoppi.emory.edu/ (accessed on 3 September 2022)	A comprehensive PPI network database concerning cancer;	This tool is used in cancer research to provide genetic, pharmacological, clinical, and structural data and combine them with the network of cancer-associated PPIs experimentally found in tumor cells	[38,39]
BioGRID	https://thebiogrid.org/ (accessed on 3 September 2022)	A comprehensive repository of PPI data providing information on their druggability;	A useful tool to study oncoprotein-chemical compound associations, based on experimental data (freely available in a variety of standardized formats)	[58]
IntAct	https://www.ebi.ac.uk/intact/home (accessed on 3 September 2022)	This is an open-source PPI database. IntAct data are organized in three clusters of information (proteomes, datasets, and mutations) that simplify the search for database entries.	A useful tool to analyze current experimentally derived PPI data from the published scientific literature; it also offers free tools for integration and analysis purposes	[59]

**Table 2 cells-11-03739-t002:** List of relevant in silico tools used to predict and analyze SLiMs involved in PPI networks.

Name of In Silico Resource	Website	Description	Technical Advantages	Refs
PIPE (Protein–Protein Interaction Prediction Engine)	https://pipe.rcc.fsu.edu/ (accessed on 4 September 2022)	This algorithm predicts the binding sites involved in PPIs based on query protein sequences and a database of known binary interaction data. The outcome is a three-dimensional graph where the peaks signify a high co-occurrence of the corresponding sequences among known interacting proteins	A useful tool to identify PPI consensus motifs. The PIPE method is based on re-occurrences of peptide sequences that mediate a large number of PPIs.	[62]
MnM (Minimotif Miner database)	http://minimotifminer.org or http://mnm.engr.uconn.edu (accessed on 4 September 2022)	This comprehensive database reports over 300,000 functional SLiMs in protein queries.	A database designed to improve prediction accuracy, allowing users to search for SLiMs with a set of false-positive filters and linear regression scoring.	[63]
LMPID (Linear Motif Mediated Protein Interaction Database)	http://bicresources.jcbose.ac.in/ssaha4/lmpid (accessed on 4 September 2022)	This manually curated database reports experimentally validated data about SLiM-mediated PPIs from any organism. It contains comprehensive information about 1762 unique SLiMs mediating 2215 PPIs among 1187 bait and 559 prey proteins.	This tool is mainly used to improve knowledge of the patterns of SLiMs binding to a specific domain and to formulate PPI inhibitors/modulators of interest.	[64]
ADAN (protein-protein interAction of moDular domAiN database)	https://adan-embl.ibmc.umh.es/ (accessed on 4 September 2022)	This manually integrated and curated database is used for the prediction of PPI-mediating SLiMs. It currently contains 3505 entries comprising structural and functional SLiM information (biochemical data, sequence files, and alignments), which is cross-referenced to other databases. The in silico prediction method is based on position-specific scoring matrices.	A useful tool to predict exact SLiMs and the best ligand and putative binding partner candidates of a protein of interest.	[65]
ELM (Eukaryotic Linear Motif)	http://elm.eu.org/ (accessed on 4 September 2022)	This manually curated platform contains different types of experimentally validated SLiM data from current literature. The classification of ELM entries is based on motif type, functional site, and ELM class. The ELM class is a specific list of experimentally validated SLiMs matching the examined query sequence.	A very versatile resource that is useful for various purposes in SLiM-related studies. It provides both a database of annotated SLiM data and an exploratory tool to predict them.	[15]
SLiMAN (is a recent database)	https://sliman.cbs.cnrs.fr (accessed on 4 September 2022)	This web server contains complementary information from the Uniprot (https://www.uniprot.org/; accessed on 4 September 2022), ELM (http://elm.eu.org/; accessed on 4 September 2022), IUpred2 (https://iupred2a.elte.hu/; accessed on 4 September 2022), BioGrid (https://thebiogrid.org/; accessed on 4 September 2022), and PhosphoSitePlus (https://www.phosphosite.org/; accessed on 4 September 2022) databases. These databases have been integrated to provide a comprehensive analysis of SLiM sequences (annotated in the ELM database), motif disorder scores (annotated in the IUpred2 server), predicted and experimentally validated PPIs (annotated in the BioGrid database), and PTMs (annotated in the Uniprot and PhosphoSitePlus databases).	A useful tool designed to overcome the various limitations related to the complex characteristics intrinsic to SLiMs (i.e., their typical localization in disordered regions or loops, their short but variable length, the varying conservation of their sequence, and their slightly bent structure).	[60]

**Table 3 cells-11-03739-t003:** SLiMs found in phosphorylation sites of human GSK3β substrates that can be altered in CRC ^1^. SLiM sequences that are subject to GSK3β phosphorylation are underlined.

Uniprot Acc. #, Gene, Entry Name	SLiM Start	SLiM End	SLiM Sequence	No. of Evidence	Experimental Evidence	Refs
Q9BYG3, MKI67IP MK67I_HUMAN	227 231	234 238	LDTPEKTVDSQGPTPVCTPT EKTVDSQGPTPVCTPTFLER	3 3	Protein kinase assay; mass spectrometry; mutation analysis	[86,87]
Q92731, ESR2 ESR2_HUMAN	5	12	MDIKNSPSSLNSPSSYNCSQ	3	Inhibitor; western blotting; mutation analysis;	[88,89]
Q5JSP0, FGD3 FGD3_HUMAN	73 77	80 84	GSLKIPNRDSGIDSPSSSVA IPNRDSGIDSPSSSVAGENF	5 2	Protein kinase assay; mutation analysis; co-immunoprecipitation; alanine scanning	[90,91]
Q15797, SMAD1 SMAD1_HUMAN	199 207	206 214	PNSPGSSSSTYPHSPTSSDP STYPHSPTSSDPGSPFQMPA	4 4	Protein kinase assay; radiolabeling; mutation analysis; western blotting	[92]
Q00613, HSF1 HSF1_HUMAN	300	307	LVRVKEEPPSPPQSPRVEEA	2	Protein kinase assay; mutation analysis;	[93,94]
P98174, FGD1 FGD1_HUMAN	280	287	DGEKVPNRDSGIDSISSPSN	1	Inhibitor	[95,96]
P84022, SMAD3 SMAD3_HUMAN	201	208	QMNHSMDAGSPNLSPNPMSP	3	Knock out; mutation analysis; protein kinase assay	[72,97]
P54252 ATXN3 ATX3_HUMAN	253	260	ADLRRAIQLSMQGSSRNISQ	2	Protein kinase assay; mutation analysis	[98,99]
P35222, CTNNB1 CTNB1_HUMAN	30	37	HWQQQSYLDSGIHSGATTTA	4	Protein kinase assay; inhibitor	[100,101,102]
P24864, CCNE1 CCNE1_HUMAN	373 388 392	380 395 399	EQNRASPLPSGLLTPPQSGKEQ EQNRASPLPSGLLTPPQSGK ASPLPSGLLTPPQSGKKQSS	4 4 1	Protein kinase assay; mutation analysis; two-dimensional phosphopeptide mapping	[83]
P05412, JUN JUN_HUMAN	236	243	QTVPEMPGETPPLSPIDMES	1	Two-dimensional phosphopeptide mapping	[103]
P04637, TP53 P53_HUMAN	30	37	KLLPENNVLSPLPSQAMDDL	2	Protein kinase assay; mutation analysis	[104]
P01106, MYC MYC_HUMAN	55	62	IWKKFELLPTPPLSPSRRSG	2	Co-immunoprecipitation; western blotting	[105,106]
O95863, SNAI1 SNAI1_HUMAN	93	100	ELTSLSDEDSGKGSQPPSPP	3	Co-immunoprecipitation; western blotting; alanine scanning	[107,108]
O95644, NFATC1 NFAC1_HUMAN	238 287	245 294	GSPRHSPSTSPRASVTEESW HSPTPSPHGSPRVSVTDDSW	2 2	Protein kinase assay; mutation analysis	[109,110]

^1^ Source data: ELM database, http://elm.eu.org/elms/MOD_GSK3_1.html; accessed on 5 September 2022.

**Table 4 cells-11-03739-t004:** Clinical trials evaluating short peptides as a treatment for CRC. Data gathered from https://clinicaltrials.gov (accessed on 5 September 2022).

Trial Status	Trial ID	Title	Treatment (s)	Result Availability
Completed	NCT00019331	Vaccine Therapy Plus Biological Therapy in Treating Adults with Metastatic Solid Tumors	Biological: Ras peptide cancer vaccine Biological: aldesleukin Biological: sargramostim Drug: DetoxPC	Not available
	NCT00098943	NGR-TNF in Treating Patients with Advanced Solid Tumors	Biological: CNGRC peptide-TNF alpha conjugate	Not available
	NCT00020267	Vaccine Therapy in Treating Patients with Metastatic Cancer	Drug: interleukin-2 Drug: MAGE-12 peptide vaccine Drug: montanide ISA-51	Not available
	NCT01364844	Safety and Tolerability of DS-7423 in Subjects with Advanced Solid Malignant Tumors	Drug: DS-7423	Not available
	NCT00841191	A Safety, Efficacy, and Pharmacokinetic Study of Siltuximab (CNTO 328) in Participants with Solid Tumors	Drug: CNTO 328; anti-interleukin-6 monoclonal antibody	Not available
Terminated	NCT00091286	Vaccine Therapy in Treating Patients with Stage IIB, Stage III, or Stage IV Colorectal Cancer	Biological: HER-2-neu, CEA peptides, GM-CSF, montanide ISA-51 vaccine	Not available
	NCT00677612	Histocompatibility Leukocyte Antigen (HLA)-A*0201 Restricted Peptide Vaccine Therapy in Patients with Colorectal Cancer	Biological: VEGFR1 and VEGFR2	Not available
	NCT00677287	Histocompatibility Leukocyte Antigen (HLA)-A*2402 Restricted Peptide Vaccine Therapy in Patients with Colorectal Cancer	Biological: RNF43, TOMM34, VEGFR1 and VEGFR2	Not available
	NCT02300922	Pretargeted Radioimmunotherapy in Metastatic Colorectal Cancer	Drug: antibody TF2 Drug: 90-Y-IMP-288 Drug: 111-In-IMP-288	Not available
	NCT03724253	[68Ga]-NeoBOMB1 Imaging in Patients with Malignancies Known to Overexpress Gastrin Releasing Peptide Receptor (GRPR)	Drug: [68Ga]-NeoBOMB1	Available
	NCT00012246	Vaccine Therapy in Treating Patients with Cancer of the Gastrointestinal Tract	Biological: carcinoembryonic antigen peptide 1–6D Biological: incomplete Freund’s adjuvant Biological: sargramostim	Not available
	NCT01925274	A Study Of PF-05212384 Plus Irinotecan vs. Cetuximab Plus Irinotecan in Patients with KRAS And NRAS Wild Type Metastatic Colorectal Cancer	Drug: PF-05212384 Drug: irinotecan Drug: cetuximab Drug: irinotecan	Available
	NCT03148119	Study of QRH-882260 Heptapeptide Application in the Colon	Drug: QRH-882260 heptapeptide Device: scanning fiber endoscope	Not available

## Data Availability

Not applicable.

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
