# Peer review of "Short Linear Motifs in Colorectal Cancer Interactome and Tumorigenesis"

_cells, 2022, doi:10.3390/cells11233739_

Round 1

Reviewer 1 Report

The manuscript entitled “Short linear motifs in colorectal cancer interactome and tumorigenesis” is a review focused on description of methods and approaches for discovering short linear motifs (SLiMs) in protein-protein interactions networks and their association with colorectal cancer. This study is of importance for biomedical and clinical researchers. In general, this article is well-written and organized and contains about 40% of relevant references published last 5 years. It contains one Figure and two Tables to illustrate the main findings and concepts in this field. However, there are some remarks and recommendations that can help to improve the quality of the manuscript.

1.      The authors did not provide any short information regarding the association of SLiMs with other types of cancer (for example, hepatocellular carcinoma and breast cancer) and diseases.

2.      It will be useful if authors consider recent articles devoted to short peptide motifs revealed in different proteins and associated with their common functions. For example, a recent review by Sologova et al., 2022 (doi: 10.3390/metabo12050464) and related articles on oncofetal proteins by Moldogazieva and co-authors.

3.      What are advantages and disadvantages of Web-resources and approaches to discover SLiMs in PPIs described in the manuscript? Could authors compare them shortly? 

4.      What are the difficulties in prediction of SLiMs and their validation? 

5.      It would be nice if additional figures were added.

6.      Please, replace “tridimensional” with “three-dimensional”.

 As a whole, English language style is perfect and the manuscript is well-readable. However, a more critical reviewing of in silico methods and approaches would be appropriate.

Author Response

Reviewer 1

The manuscript entitled “Short linear motifs in colorectal cancer interactome and tumorigenesis” is a review focused on description of methods and approaches for discovering short linear motifs (SLiMs) in protein-protein interactions networks and their association with colorectal cancer. This study is of importance for biomedical and clinical researchers. In general, this article is well-written and organized and contains about 40% of relevant references published last 5 years. It contains one Figure and two Tables to illustrate the main findings and concepts in this field. However, there are some remarks and recommendations that can help to improve the quality of the manuscript.

   1.The authors did not provide any short information regarding the association of SLiMs with other types of cancer (for example, hepatocellular carcinoma and breast cancer) and diseases.

We are grateful to the reviewer for this suggestion. In this amended version of the manuscript, we divided the introduction into a short introductory paragraph and a separate section specifically focused on SLiMs, which we entitled “Short liner motifs: general features and emerging role in cell biology and cancer”. This section reports additional evidence of SLiM biological relevance in other types of cancer.

  1. It will be useful if authors consider recent articles devoted to short peptide motifs revealed in different proteins and associated with their common functions. For example, a recent review by Sologova et al., 2022 (DOI: 10.3390/metabo12050464) and related articles on oncofetal proteins by Moldogazieva and co-authors.

We thank the reviewer for this useful observation. We considered the suggested articles and discussed this point in the new section entitled “Short liner motifs: general features and emerging role in cell biology and cancer”.

  1. What are advantages and disadvantages of Web-resources and approaches to discover SLiMs in PPIs described in the manuscript? Could authors compare them shortly?

We thank the reviewer for raising this point. In agreement with the suggestions from both reviewers, in this amended version of the manuscript, we revised the sections on current in silico tools for the analysis of PPI networks and SLiMs. In particular, the technical information about the tools has been reorganized and summarized in the new Table 1 (tools for PPI network analysis) and Table 2 (tools for SLiM analysis). In both tables, was added a column that highlights the advantages of each tool. In addition, we briefly discussed these in silico approaches for SLiMs analysis and prediction in PPIs.

  1. What are the difficulties in prediction of SLiMs and their validation?

We thank the reviewer for this interesting question. In this revised version of the manuscript, we included considerations on the limitations of in silico SLiM prediction in the section entitled “In silico approaches and tools to characterize SLiMs in PPI networks”.

  1. It would be nice if additional figures were added.

We agree with the reviewer that graphical elements are a useful way to visualize/summarize data. The present manuscript contains a figure illustrating a SLiM-based method for identifying novel interactors of an oncoprotein of interest (Figure 1) and a figure depicting the rationale behind the use of SLiM-based anticancer drugs (graphical abstract). Besides revising Figure 1, in this amended version of the manuscript, we opted for adding two new tables related to the in silico tools for prediction and analysis of PPIs and SLiMs (Table 1 and Table 2, respectively), which we believed would be more helpful to summarize the technical details reported in Sections 3 and 4.

  1. Please, replace “tridimensional” with “three-dimensional”. As a whole, English language style is perfect and the manuscript is well-readable.

We thank the reviewer for his/her general comment about English language style and for spotting this typo, which we corrected in this amended version of the manuscript. 

  1. However, a more critical reviewing of in silico methods and approaches would be appropriate.

We thank the reviewer for raising this point, and we agree with him/her that the sections on in silico methods would benefit from a general revision. To this aim, in this amended version of the manuscript, we summarized the information about the in silico tools for prediction and analysis of PPIs and SLiMs into two new tables (Table 1 and Table 2, respectively) and we critically reviewed their advantages and limitations.

Reviewer 2 Report

The review entitled ‘Short linear motifs in colorectal cancer interactome and tumorigenesis’ focused on Short linear motifs and investigated its utilization in CRC interactome and tumorigenesis. The topic is interesting and novel. However, some issues should be addressed. 1. The abstract should be re-written. It is suggested to focus on the findings for a review, not the steps. Importantly, the authors should demonstrate the strategy for searching the articles. 2. Please add the first section as ‘The introduction of Short linear motifs’, which should be provide a comprehensive introduction of this concept. Then, the followed section of ‘in silico methods’ should be simplified. The list of different online dataset is not suitable for introducing this part. It is suggested to delete such parts and summary as a table for readers. Similar problem was also found in next part. 3. For the section ‘SLiMs in CRC molecular networks and tumorigenesis’ and ‘Potential small-molecule anticancer drugs based on SLiMs and short peptides to interfere with CRC interactome and tumorigenesis: where do we stand?’, too long to get useful information. It is suggested to subtitle these parts based on signal pathway, or cancer hallmarks or other aspects. 4. Figure 1 is too unclear. 5. For Table 2, it is suggested to focus on CRC, not all the types of cancers involved in the clinical trials, which may confuse the readers. So please reconstruct the table titles and its columns. 6. Last not the least, the English writing should be polished by a native English-speaker with biological background. Too many paragraphs only have one sentence without conclusion or summarization.

Author Response

Reviewer 2

The review entitled ‘Short linear motifs in colorectal cancer interactome and tumorigenesis’ focused on Short linear motifs and investigated its utilization in CRC interactome and tumorigenesis. The topic is interesting and novel. However, some issues should be addressed.

1.The abstract should be re-written. It is suggested to focus on the findings for a review, not the steps.

We are grateful to the reviewer for this comment, and we agree with his/her suggestion. Thus, in this amended version of the manuscript, we revised the abstract by focusing on the emerging biological relevance of SLiMs in cancer disease, especially in CRC.

  1. Importantly, the authors should demonstrate the strategy for searching the articles.

Since this is an emerging topic, the strategy for searching relevant articles was based on querying the PubMed database for recent studies involving short linear motifs and protein-protein interactions. Then, we filtered the results by using the keywords “colorectal cancer” and “CRC”.

  1. Please add the first section as ‘The introduction of Short linear motifs’, which should be provide a comprehensive introduction of this concept. Then, the followed section of ‘in silico methods’ should be simplified. The list of different online dataset is not suitable for introducing this part. It is suggested to delete such parts and summary as a table for readers. Similar problem was also found in next part.

We agree with the reviewer that these sections would benefit from a general reformulation. Thus, in this amended version of the manuscript, we divided the introduction by adding a separate section specifically focused on SLiMs entitled “Short liner motifs: general features and emerging role in cell biology and cancer”. Moreover, we summarized the information about the in silico tools for prediction and analysis of PPIs and SLiMs into two new tables (Table 1 and Table 2, respectively).

  1. For the section ‘SLiMs in CRC molecular networks and tumorigenesis’ and ‘Potential small-molecule anticancer drugs based on SLiMs and short peptides to interfere with CRC interactome and tumorigenesis: where do we stand?’, too long to get useful information. It is suggested to subtitle these parts based on signal pathway, or cancer hallmarks or other aspects.

We thank the reviewer for this useful suggestion. In this amended version of the manuscript, we split these sections into subsections to increase overall readability, as detailed below:

Section 5 SLiMs in CRC molecular networks

5.1 SLiMs in CRC signaling pathways and tumorigenesis

5.2 SLiMs in CRC hallmarks: a case study

5.3 SLiMs in CRC-related microbiome

Section 6. Potential small-molecule anticancer drugs based on SLiMs and short peptides in CRC: where do we stand?

6.1 Pharmacological suitability of SLiMs and short peptides as anticancer drugs

6.2 Short peptides as potential anti-CRC drugs

6.3 Short peptides in clinical studies

  1. Figure 1 is too unclear.

We thank the reviewer for this observation. In this amended version of the manuscript, we revised Figure 1 and we better described in the figure legend the workflow steps of the depicted SLiM-based method for identifying novel interactors of an oncoprotein of interest.

  1. For Table 2, it is suggested to focus on CRC, not all the types of cancers involved in the clinical trials, which may confuse the readers. So please reconstruct the table titles and its columns.

We are grateful to the reviewer for this comment, and we agree with his/her suggestion. In this amended version of the manuscript, we revised old Table 2 (now Table 4) by deleting the column detailing tumor types.

  1. Last not the least, the English writing should be polished by a native English-speaker with biological background. Too many paragraphs only have one sentence without conclusion or summarization.

We thank the reviewer for this comment. We agree that the text would benefit from the addition of a conclusion and/or summarization at the end of the different sections of the manuscript. Furthermore, the manuscript was carefully revised to improve grammar and readability.

Round 2

Reviewer 2 Report

The manuscript is improved. It is still suggested to reduce Table 4.